# Digital Surveillance to Identify California Alternative and Emerging Tobacco Industry Policy Influence and Mobilization on Facebook

**DOI:** 10.3390/ijerph182111150

**Published:** 2021-10-23

**Authors:** Qing Xu, Joshua Yang, Michael R. Haupt, Mingxiang Cai, Matthew C. Nali, Tim K. Mackey

**Affiliations:** 1Global Health Policy and Data Institute, San Diego, CA 92121, USA; qing.xu@s-3.io (Q.X.); mhaupt@ucsd.edu (M.R.H.); mcai@eng.ucsd.edu (M.C.); mnali@ucsd.edu (M.C.N.); 2S-3 Research, LLC, San Diego, CA 92121, USA; 3Department of Public Health, California State University, Fullerton, CA 92834, USA; jsyang@Fullerton.edu; 4Department of Cognitive Science, University of California, San Diego, CA 92093, USA; 5Global Health Program, Department of Anthropology, University of California, San Diego, CA 92093, USA

**Keywords:** tobacco control, electronic cigarettes, social media, Facebook, public policy

## Abstract

Growing popularity of electronic nicotine-delivery systems (ENDS) has coincided with a need to strengthen tobacco-control policy. In response, the ENDS industry has taken actions to mobilize against public health measures, including coordination on social media platforms. To explore this phenomenon, data mining was used to collect public posts on two Facebook public group pages: the California Consumer Advocates for Smoke Free Alternatives Association (CCASAA) and the community page of the Northern California Chapter of SFATA (NC-SFATA). Posts were manually annotated to characterize themes associated with industry political interference and user interaction. We collected 288 posts from the NC-SFATA and 411 posts from CCASAA. A total of 522 (74.7%) posts were categorized as a form of political interference, with 339 posts (64.9%) from CCASAA and 183 posts (35.1%) from NC-SFATA. We identified three different categories of policy interference-related posts: (1) providing updates on ENDS-related policy at the federal, state, and local levels; (2) sharing opinions about ENDS-related policies; (3) posts related to scientific information related to vaping; and (4) calls to action to mobilize against tobacco/ENDS policies. Our findings indicate that pro-tobacco social media communities on Facebook, driven by strategic activities of trade associations and their members, may act as focal points for anti-policy information dissemination, grass-roots mobilization, and industry coordination that needs further research.

## 1. Introduction

The growing popularity of electronic nicotine-delivery systems (ENDS) has resulted in the expansion of the tobacco industry and new concerns about aggressive marketing, underage, and young adult uptake, and the growing presence of industry interference (i.e., various tactics used to defeat, weaken or delay public health measures meant to reduce tobacco use) in state and federal tobacco-control policies [1,2,3,4,5,6,7,8,9] The emergence of ENDS, including different types of e-cigarettes and other emerging and alternative products (e.g., Heated Tobacco Products (HTPs)) along with their differential pricing, represent new market entries with their own unique health risks, different channels of product marketing and sales, and the presence of new and old tobacco industry actors who promote their use and advocate for their protection from public policymaking [10,11,12,13]. However, there have been signs of a slowdown in ENDS sales, emphasizing the importance of continuing to strengthen state and federal tobacco-control efforts that have focused on instituting new tobacco/ENDS product taxes, raising the minimum age of sale, and sales bans on certain products (e.g., flavored ENDS products), all aimed at reducing uptake and appeal [8,14].

Active in enacting tobacco-control policy efforts is the state of California, a focal point in efforts to regulate tobacco and nicotine products via enactment of progressive policies that now subject ENDS to existing anti-tobacco laws (SB X2-5, Senate Bill No.5 Electronic ciarattes), raised the minimum purchasing age to 21 (SB X2-7, Senate Bill No.7 Tobacco products: minimum legal age), and imposed additional taxes on cigarettes and ENDS (Proposition 56). In response, the tobacco and ENDS industry has become aggressive in mobilizing opposition to these policies, using its expansive and increasingly diverse network of manufacturers, trade associations, and tobacco and vaping user communities, including through traditional political advocacy activities (such as lobbying), coordination across industry stakeholders (efforts by trade associations), and grass-roots activities by organizing opposition to tobacco-control measures through live and online campaigns with vaping constituency groups [8,15].

Facebook, as one of the world’s most popular social media platforms, now serves more than 2.7 billion monthly active users globally, and has been used for political mobilization [16,17,18,19]. As a global social networking site, Facebook allows individual users and organizations to create profiles and group pages, share information, send event invitations, communicate through open and direct messages, and has a number of embedded web- and mobile-based applications. The combination of these functions makes Facebook a powerful platform for gathering communities, market products using influencers to promote tobacco brands and lifestyles (though Facebook now restricts sponsored ads and influencers for tobacco products), and also represents an important tool for the tobacco/ENDS industry to activate and mobilize opposition to tobacco-control laws [20,21,22,23].

A number of published studies have used social media to: identify trends associated with ENDS-related attitudes and behaviors; understand opinions about product harms and benefits; explore ENDS marketing, sales, and pricing; develop methodologies to categorize and analyze ENDS-related social media content; and identify geographic locations where people use ENDS [2,3,21,24,25,26,27,28,29,30,31,32,33,34] However, only a handful of studies have specifically assessed how social media can influence tobacco-control public health campaigns and public policies, including studies that have examined how the industry attempts to influence the outcome and implementation of tobacco-control policy and campaigns [4,35,36]. This study builds on this prior research by further identifying how the ENDS industry specifically coordinates through industry trade groups on Facebook, important as few if any studies have examined industry influence on the platform (prior studies have primarily focused on Twitter). The specific objective of the study is to identify and characterize strategic approaches used by these online communities aimed at influencing the outcome of California and federal tobacco-control legislation. Findings can help in the design of counter-marketing campaigns and advocacy activities aimed to at strengthening tobacco-control policy efforts.

## 2. Materials and Methods

### 2.1. Data Collection

In order to observe interactions involving the California ENDS industry—including manufacturers, trade associations, retailers, and vaping product users—we selected two politically active ENDS advocacy associations who operate in California and have Facebook pages for the purposes of data collection, including: (1) the “California Consumer Advocates for Smoke-Free Alternatives Association” (CCASAA)—a Facebook public group page organized by the Consumer Advocates for Smoke-free Alternatives Association (CASAA); and (2) the “Northern California Chapter of Smoke-Free Alternatives Trade Association” (NC-SFATA)—a Facebook community page created by NC-SFATA as the official page for its chapter. CCASSA states that it is a consumer advocacy nonprofit organization dedicated to empowering vaping consumers and representing their interests in legislative, policy making, and rulemaking areas [37], and NC-SFATA states that it is a vaping products trade organization dedicated to representing the interests of small- to mid-sized businesses by engaging with political decision-makers via advocacy [38]. We used data mining approaches in the computer programming language Python to collect all publicly available Facebook user-generated posts from these pages, which collected data retrospectively for all posts still available and posted to these accounts up until 14 July 2020. No private messages or non-public information from these pages was collected.

The unit of analysis for this study was a Facebook post or interaction associated with the CCASSA or NC-SFATA page. In order to characterize the interaction between those who posted messages to these pages and other Facebook users who interacted with these posts, we collected the following information from each post: (1) text of the Facebook post; (2) extended resource associated with the post (shared URLs/hyperlinks to external sites); (3) the number and content of any comments made to the post by other users; (4) the number of shares of the post; and (5) the number and type of reactions to the post (e.g., likes and emoji reactions). We also examined publicly available metadata of users to determine if there were any professional or employment affiliations claimed with the tobacco/ENDS industry.

### 2.2. Data Analysis

#### 2.2.1. Content Analysis

To classify the content of Facebook posts, a general inductive approach was utilized to code the textual data [39]. Facebook posts were reviewed twice by the second author (JSY) and notes were taken on general themes of posts from which an initial code list was created. Formal coding of text data was conducted with codes refined and subcodes created. A final coded dataset was reviewed by the first author (QX) and differences in code definitions and application were reconciled by the first and second coder. Inductively derived codes (see Appendix A) were then included in the analysis if they included content categorized as an industry political interference strategy [40]. A binary coding scheme was created to include codes that met one of the political strategies used by the tobacco industry as defined by Savell et al., including: (1) information (including direct and indirect lobbying, shaping the evidence base, and establishing industry/policymaker collaboration); (2) constituency building (forming alliances with other sectors/busines/trade organizations, media advocacy, organizations, and formation of alliances with unions/civil society organization/consumers/the public, and collaboration between companies); (3) policy substitution (developing or promoting voluntary regulation, alternative regulation, or non-regulatory initiatives); (4) legal efforts (pre-emption, and threat of legal action); (5) constituency fragmentation and destabilization (preventing emergence of, neutralizing, and/or discrediting potential opponents); and (6) financial incentives (current or future employment, gifts, entertainment, or other financial inducement) [40].

Posts were excluded from the analysis if they did not fit one of these categories, which comprised a total of 177 posts. Included codes were then assigned one of the six types of political interference. Assigning of codes to political interference type was conducted by the first author and reviewed for appropriateness by two additional authors (JSY, and TM). Data coding was conducted using qualitative software Atlas.ti version 8 (ATLAS ti, Berlin, Germany). The first (QX) and second (JY) authors coded all posts independently and achieved a high inter-coder reliability for post signal coding (kappa = 93.46). A final coded data set was reviewed by the first, second, and last author (TKM) to assess if any differences in code definitions and application occurred and reconciled differences by reaching consensus on the correct classification.

#### 2.2.2. Engagement Analysis

In order to characterize the potential dissemination of these posts through online communities, we also developed a typology to measure Facebook user post engagement. This was accomplished by collecting and analyzing metadata from each post included for analysis and from associated user interactions (e.g., number of text comments per post, and the number of likes and other emoji reactions), with different engagement typologies plotted on a continuum of intensity, from low to high activity adopted from an existing typology of social media engagement behavior (SMEB) [41] specific to Facebook data (see Appendix A and descriptions in Table 1). We also examined publicly available Facebook profile metadata from users who posted messages to ascertain if any had employment or affiliations with the tobacco/ENDS industry. From this data, we calculated a user interaction engagement ratio per post as well as the percentage of policy and political interference-related posts created by users who were affiliated with the vaping industry.

## 3. Results

We collected a total 699 unique Facebook posts during the study period. Of the total posts, 411 (58.8%) were collected from the public group page for CCASAA, and 288 (41.2%) from the community page for NC-SFATA. The oldest post collected from CCASAA was from November 2017 and the most recent from June 2020; with the oldest post from NC-SFATA from August 2014 and the most recent from February 2017. Tobacco/ENDS control policies specifically mentioned in these posts included numerous local ordinances and legislation in California and federally introduced, discussed, and/or passed legislative actions (see full list in Appendix A).

### 3.1. Content Analysis

A total of 522 (74.7%) posts were categorized as a form of political interference based on Savell et al. political interference categories, with 339 posts (64.9% of all identified posts) from CCASAA and 183 posts (35.1%) from NC-SFATA. Based on our qualitative analysis, four main types of political interference were detected: (i) information—providing or spreading false or misleading information by understating the health benefits of a proposed policy and overstating its social and economic consequences; (ii) policy substitution—providing policy substitutes, where alternative policies are developed as a substitute for proposed policies; (iii) constituency building—gaining the support of other sectors, organizations, or individuals in order to give the impression of a larger support base for the industry position; and (iv) legal—using or threatening legal action against proposed policies [40]. Among these, there were four main inductively derived categories of political interference: (1) providing updates on e-cigarette-related policy at the federal, state, and local levels (example: “Heads up, Belmont, CA! Flavor ban being considered September 25th!”; meeting the criteria for “constituent building”); (2) sharing opinions about e-cigarette-related policies (example: “Mike Males, senior researcher for the Center on Juvenile and Criminal Justice, drops a fact bomb on the effectiveness of the Tobacco 21 law in California”; meeting the criteria for “policy substitution”); (3) posts related to scientific information related to vaping (example: “There’s NO science supporting OUTDOOR bans & misleads public about vaping risks!”; meeting the criteria for “information”); and (4) calls to action on tobacco/e-cigarette policies (example: “OAKLAND, CA: (CALL TO ACTION!) Stop a TOTAL FLAVOR BAN!”]; meeting criteria for “information”). Additional example posts for each of these political interference categories are included in Table 2.

We calculated the number of posts for each political influence theme on both pages, which included overlapping themes in the same post, and detected 198 (37.9%) information posts (CCASAA: 115, NC-SFATA: 83), 49 (9.4%) policy-substitution posts (CCASAA: 32, NC-SFATA: 17); 351 (67.2%) constituency-building posts (CCASAA: 288, NC-SFATA: 63); and 4 (0.8%) legal posts (CCASAA: 3, NC-SFATA: 1). Overall, the “Constituency Building” category (55.2%), and the “Information” category in CCASAA (22.0%) had the largest proportion of the total volume of industry interference-related posts, followed by the “Information” category (15.9%), and the “Constituency Building” category in NC-SFATA (12.1%). In contrast, the “Policy Substitution” category was low in both the CCASAA (6.1%) and NC-SFATA page (3.3%), and the lowest was the “Legal” category in CCASAA (0.6%) and NC-SFATA (0.1%) (See Figure 1 and Appendix A).

For posts that did not meet the binary coding classification requirement as per Savell et al.’s framework, we nevertheless detected messages that could be interpreted as attempting to indirectly influence public perception and impact the outcome of tobacco-control initiatives. This included posts that shared information or quoted language from published studies purporting the health benefits of vaping and promotional posts for vaping products that also included policy-related comments from other users. We also noticed that 54 posts (10.34%) shared articles or reports that questioned the veracity of scientific studies (including doubts about study methodology or reliability of data) related to the negative health impacts of ENDS and vaping, including its association with cancer, heart disease, and other risk factors. Comments also shared other media including interviews and articles that claimed vaping bans caused a large number of vapers to transition smoking combustible products. Of these posts, 19 (35.19%) were action oriented, calling for the public to take action based on evidence they claimed supported opposition of enactment of ENDS-related policies.

### 3.2. User Interaction Analysis

This study also measured interaction between users. However, it is important to note that the CCASAA and NC-SFATA pages are different types of Facebook pages (e.g., public group page versus community page.) A major difference in these pages is that the community page has one or more authorized page managers that can create posts, while in a public group page, every group member can create posts without any form of moderation. In total we found 2337 user engagements via comments to posts on the two Facebook pages (1229 comments on CCASAA and 1108 comments on NC-SFATA.) Hence, despite employing different levels of moderation for user engagement by restricting who can create posts, both pages generated over 1000 user interactions via comments. User metadata analysis found that the community NC-SFATA page had a higher average user engagement ratio (6.05) for each post compared to the public CCASAA group page ratio (3.60.) This included 2324 (99.44%) positive engagements and 13 (0.56%) negative engagements per the SMEB scale. In the positive engagement category, 1292 fit the typology of “support” level and 1032 with “agree”. In the much smaller group of negative engagements, we found 7 “oppose” and 6 “disagree” level engagements, with both of these categories of negative engagement detected on the CCASSA page (See Figure 2). In categorizing “support” level engagement, writing positive comments in reaction to another user’s post and sharing the post were considered agreeing with the content of the post and supporting its broader online dissemination.

### 3.3. Industry Affiliations

Based on analysis of the self-reported profile metadata of users and account managers on the Facebook pages reviewed, we were able to conduct a descriptive analysis of possible industry affiliations. For the CCASAA page, only 28 unique user accounts from those who created posts and 53 users that interacted with these posts included occupation affiliations in their profile. Six users who posted messages and 7 users who interacted with these messages had prior work experience or claimed to be currently working in the ENDS industry. This included individuals who were employees of ENDS manufacturers, retail shops, trade associations, online vendors, and wholesalers. However, upon further examination it was determined that these industry affiliated users contributed more than half (52.8% *n* = 179) of all posts reviewed and 3.2% (*n* = 39) of all observed interaction behavior with other users on the CCASAA page. On the NC-SFATA page, only page managers are allowed to create post, though as a public page, any Facebook user (including those who are not members of the page), can interact with posts. Hence, among all 151 active users that had an interaction on the page, 62 had occupation affiliations indicating prior or current work in the ENDS industry, with these users contributing 9.0% (*n* = 100) of all comments.

## 4. Discussion

Our review of user posts generated from two politically active California-based ENDS trade association Facebook pages found that close to three-quarters of all posts communicated messages that were categorized as a form of political interference. These posts generated 2337 user engagements, with virtually all of these engagements positive in the context of supporting political interference themes and other pro-tobacco messaging. Though only in the thousands in overall volume, these posts and the user engagement they generated, likely led to more widespread dissemination of pro-tobacco and anti-legislation messages across the Facebook platform, potentially among users who are not members or who post to these pro-ENDS groups.

Unsurprisingly, Facebook users with tobacco/ENDS industry affiliations were active in these online discussions, including on the CCASAA page where we observed that more than half of the posts generated were contributed by accounts with clear industry affiliations. We also observed increased engagement on both the CCASSA and NC-SFATA pages during periods of federal legislative activity. For example, we recorded a spike in the number of posts and total engagements during September 2019, beginning the week following the introduction of H.R.4249 on 9 September 2019, a possible indication of mobilization efforts to defeat the same tobacco-control policy measure. We also observed that a key leader within the NC-SFATA group was involved in the CCASAA page, which could indicate potential coordination between groups. Hence, these pages appear to have the central purpose of mobilizing, disseminating information, and coordinating activities that constitute as digital industry interference in the policymaking process of both state and federal tobacco-control legislation.

More specifically, operators of these industry-driven Facebook community groups and public pages used several digital strategies in an attempt to influence public perception and coordinate efforts of their online constituents. These activities included acting as an online information source about tobacco-control legislative developments (including for CA local ordinances, state legislation/referendum, and federal legislation), inserting discourse and opinion against tobacco-control policy with information about policy, and possibly misrepresenting scientific information about the potential harms of ENDS and vaping. For example, posts observed included those arguing against a study that indicates that chemicals found in e-cigarettes disrupts the gut barrier and triggers inflammation in the body leading to potential harm [42]; and they also questioned the veracity of the methodology of studies that detected potentially harmful chemicals in e-cigarettes [43]. Further, these community groups make specific calls to action to mobilize against tobacco-control policies (e.g., inviting online users to political mobilization events, asking them to take direct action against tobacco-control policies, etc.). Many of these strategies overlapped in the text of a single post and, again, originated from users with clear industry affiliations.

In response to this strategic use of Facebook by the ENDS industry, public health stakeholders should expand their own efforts to mitigate and counter pro-vaping narratives, particularly if they misrepresent or include misinformation about tobacco-control legislation [44,45,46]. Importantly, in this study we did not observe any pro-public health or fact-checking of these claims by other Facebook users interacting with these pages or their posts. Hence, California public health and tobacco-control advocates should seek to directly engage with these sources of information and their audiences on social media platforms such as Facebook, with the aim or better understanding the rationale of pro-ENDS constituents, anticipating arguments against policy implementation that may lack scientific merit, and mobilizing their own public health stakeholders to counter events driven by industry mobilization online. These efforts are particularly important given the uniqueness of the California tobacco-control regulatory landscape, with the state leading the way nationally on many progressive policies such as indoor and outdoor smoking bans, efforts to remove illegal and counterfeit vaping products from stores, and a state-wide ban on flavored tobacco and ENDS products that will be decided in a ballot referendum in 2022.

Additionally, posts on these pages appear to have an “echo chamber” effect, in that virtually all the posts exclusively present a pro-vaping narrative that is disseminated among users who are members of these pages. For Facebook pages that are open to the public, these discussions could spill over into other online communities, and in the absence of effective counter-marketing, fact-checking, and health promotion, could lead to pro-tobacco messages influencing the opinions of other users, including perceptions, knowledge, and attitudes about crucial tobacco-control policy. In order to prevent these outcomes, public health stakeholders and tobacco-control advocates should make a concerted effort to engage in these almost exclusively pro-vaping virtual communities to disseminate counter narratives in support of tobacco-control policies and initiatives aimed at curbing uptake, use, and promoting cessation.

## 5. Limitations

This study has certain limitations. First our study was limited to the Facebook platform and focused on two specific ENDS trade association pages that operate in the state of California, so results are not generalizable to national tobacco or ENDS industry political or social media activities. Future studies should conduct more widespread social listening of tobacco and ENDS industry groups, including pages associated with national ENDS trade associations and interest groups (e.g., CASSA and SFTA national chapters and additional state chapters), as well as examining the accounts of specific vaping retail outlets and stores that are members of these associations. Further, this study only reviewed publicly available information on these pages and are generally available to any online user for inspection. This explicitly does not include any content generated via private communication applications (e.g., Messenger or WhatsApp, which are both services operated and owned by Facebook), which may have contained more directed coordination communication between ENDS actors and Facebook users seeking to influence the outcome of public policy. Further, since not all users provided occupation affiliations in their profile, it is possible that this study did not capture all industry affiliations associated within online communities reviewed. Additionally, we did not conduct a formal assessment of whether accounts may have been bots instead of human users, though content from posts reviewed was generally original and not duplicative of other content, hence, not indicative of spam or bot traffic. Finally, since data was collected retrospectively, each network is also susceptible to data loss for posts that may have been deleted by users prior to our data collection taking place in July 2020.

## 6. Conclusions

The results from this study show pro-vape narratives are prevalent across online spaces that may unduly influence the policy-making process, while having the potential to threaten future tobacco legislation and implementation of policy already in place. In order to address this issue, there is a need for future digital surveillance research that characterizes messaging strategies between tobacco/ENDS trade associations and lobbying organizations in the digital sphere. While this study analyzed the content from both the CCASSA and NF SFATA groups together, future work investigating a larger sample of organizations should compare differences between messaging and narratives featured between local and national organizations. Additional work comparing content differences between trade associations, lobbying organizations, retail outlets, and manufacturers can further reveal differences in mobilization strategies and possible cross-platform coordination across different parts of the internet ecosystem. Within the context of misinformation spread about public health issues more generally (as illustrated by the current COVID-19 infodemic), a more in-depth characterization of possible misinformation narratives related to tobacco-control legislation would also provide additional detail into how ENDS organizations undermine scientific evidence to mobilize users and could also inform future counter-messaging strategies.

## Figures and Tables

**Figure 1 ijerph-18-11150-f001:**
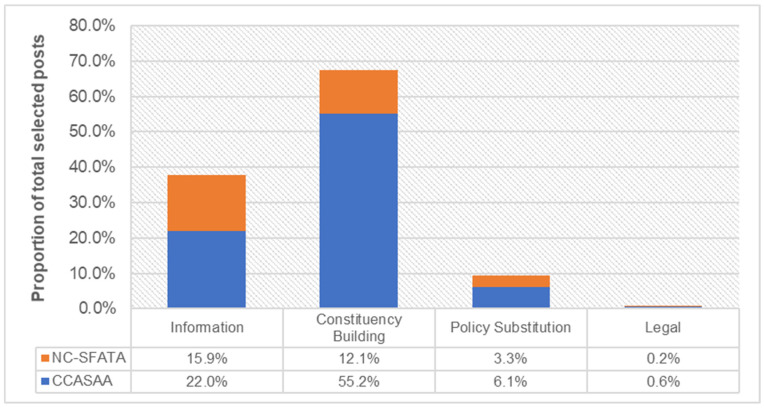
Content thematic breakdown.

**Figure 2 ijerph-18-11150-f002:**
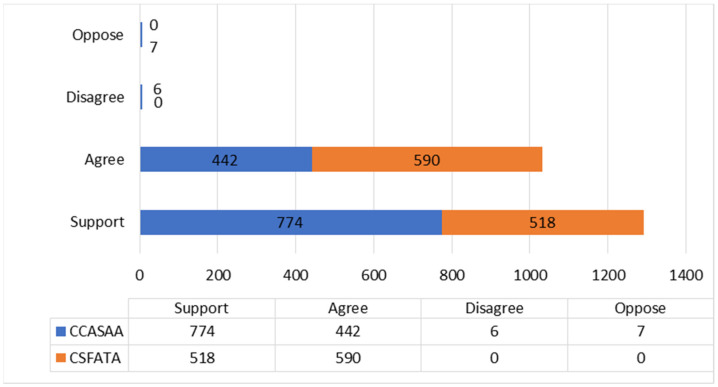
Engagement typology breakdown.

**Table 1 ijerph-18-11150-t001:** Facebook members engagement behavior construct.

	Definition	Examples
Support	The higher level of positive—active users initiate positive and active contributions to communities	Sharing signal posts and writing positive comment on posts stand for against tobacco policy
Agree	The minimum level of positive—users making positive comments without any additional language or contributions	Using “like”, “love”, and other positive emoji comments on posts stand for against tobacco policy, and using “angry”, “sad”, and other negative emoji comments on posts describe detail of tobacco-control policy.
Disagree	The minimum level of negative—users making negative comments without any additional language or contributions	Using “angry”, “sad”, and other negative emoji comments on posts stand for against tobacco policy, and using “like”, “love”, and other positive emoji comments on posts describe detail of tobacco-control policy.
Oppose	The highest level of negative—active users initiate negative contributions to the community	Writing a public censure on posts stands for against tobacco policy

**Table 2 ijerph-18-11150-t002:** Major thematic areas of political interference in Facebook posts.

Thematic	Example
Updates on e-cigarette-related policy	BEVERLY HILLS, CALIFORNIA: Lawmakers want to ban ALL tobacco sales in the city (including very low risk vapor & smoke-free products) to everyone except wealthy cigar smokers and rich hotel guests. CALIFORNIA: (HEADS UP!) SB 38, a bill to ban the sale of flavored #vaping & #smokefree products statewide, was approved in the Senate Health committee yesterday. It will be heard next in the Senate Appropriations Committee! HEADS UP FOR CALIFORNIA INDUSTRY STAKEHOLDERS. CA SB140 has been scheduled for a hearing in the Senate Health Committee on March 25th. We’re working out the details and actions required right now and will update everyone through various means in the next few days. If you want to ensure that you will receive critical and pertinent information on the actionable items, please go to http://norcal.sfata.org/ (23 June 2020) and scroll to the bottom and register for our “newsletter”. NATIONAL: (HEADS UP!) New bill (HR 293) introduced in Congress would ban sale of #vaping products and much lower risk #tobacco products across state lines (incl. online,) add excessive taxes and ban non-tobacco flavors (unless approved as NRT.)
Opinions on e-cigarette-related policies	It is funny that they are willing to ban liquid and vapes in a whole state but won’t even shut down a shop that repeatedly breaks the law and sells to minors; they just fine them. Opinion|The Ridiculous Campaign Against Vaping (https://www.politico.com/magazine/story/2019/09/18/rich-lowry-vaping-campaign-228143/ (21 October 2021)) IN THE NEWS: (OPINION) “One of the unintended consequences of targeting e-cigarette flavors may be to harm public health by limiting adults’ access to products that are proven to be helping them quit smoking cigarettes.” https://www.ocregister.com/2018/12/12/californias-plan-to-ban-vaping-flavors-would-hurt-public-health/ (21 October 2021)
Scientific information related to vaping	BAD PUBLIC HEALTH: (SAN FRANCISCO) Same DoH that ignored science & helped ban low risk #vaping products—leaving adults who smoke w/ineffective NRT, COLD TURKEY or CIGARETTES—admits harm reduction “scientifically proven more effective.” Hands out cigarettes to homeless. #QuitLying UPLAND, CA: City joins several others w/ALL public smoking/vaping ban—a law that has already led to a scary confrontation between a citizen & police officer in La Mesa, CA. There’s NO science supporting OUTDOOR bans & misleads public about vaping risks! Two recent study findings that were released to the media this month are just wild speculation, and possibly intentional misrepresentation, according to analysis by the Consumer Advocates for Smoke-free Alternatives Association (CASAA). In an as-yet-to-be-published (or peer reviewed) preliminary study, researchers looked at the saliva of just five vapers, testing for the presence of carcinogenic chemicals. Researchers found increased levels of formaldehyde, acrolein, and methylglyoxal. [816 more words] http://www.casaa.org/news/vaping-cancer-heart-disease-risk-overstated-by-recent-research/ (21 October 2021)
Calls to action tobacco/e-cigarette policy	CA—Burbank—Call To Action! PUBLIC HEARING on a proposed FLAVOR BAN Tues., Sept. 24, 6:00 p.m. Details and Take Action, here --> CALIFORNIA: (CALL TO ACTION!) SB 38, which would BAN the sale of FLAVORED #vapor and #smokefree products is scheduled for a public hearing on March 27th! IF your rep is on the Committee on Health, you can TAKE ACTION NOW! Ok everyone. Please help us help you. Call your representatives and tell them about how flavors helped you switch. If you don’t know how, please PM me and I’ll get you there. Vapers! WE NEED YOUR HELP to encourage Congress and federal regulators to reject any proposal that would ban OR limit flavored e-liquid products. Right now, FDA is considering whether to regulate flavors and you need to be heard. Read more and submit your comment: http://actnow.io/saveflavors1 (21 October 2021)

## Data Availability

The datasets used and/or analyzed during the current study are available from the corresponding author on reasonable request subject to appropriate de-identification and aggregation.

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
