# Peer review of "Digital Surveillance to Identify California Alternative and Emerging Tobacco Industry Policy Influence and Mobilization on Facebook"

_ijerph, 2021, doi:10.3390/ijerph182111150_

Round 1

Reviewer 1 Report

This paper examines public posts on two Facebook public group pages for ENDS industry support groups to determine whether there is evidence political interference among the groups.  The authors have used appropriate methods and clearly explained the results. The conclusions are well thought out and follow from the results. The limitations have been clearly addressed. I have no suggested changes to the manuscript and think this is a timely topic of research.

Author Response

REVIEWER COMMENT:  This paper examines public posts on two Facebook public group pages for ENDS industry support groups to determine whether there is evidence political interference among the groups.  The authors have used appropriate methods and clearly explained the results. The conclusions are well thought out and follow from the results. The limitations have been clearly addressed. I have no suggested changes to the manuscript and think this is a timely topic of research.

Author Response:  We thank the reviewer for the time taken to review our article and your overall support of our approach and findings.  

Reviewer 2 Report

The purpose of this article is to observe interactions involving the California ENDS (Electronic Nicotine Delivery Systems) industry - including manufacturers, trade associations, retailers, and vaping product users. To this purpose the authors have inquired Facebook pages are online by consumers associations in California. See lines 86-112.

ENDS/ENNDS (Electronic Non-Nicotine Delivery Systems) and cigarettes are substitutes, with higher cigarette prices being associated with increased ENDS/ENNDS sales. Therefore, differential tax policies based on product type could lead to substitution between different types of ENDS/ENNDS and between ENDS/ENNDS and cigarettes (see online: https://escholarship.org/content/qt2f65f2j5/qt2f65f2j5.pdf).

ENDS are the subject of a public health dispute among bona fide tobacco-control advocates that has become more divisive as their use has increased. Whereas some experts welcome ENDS as a pathway to the reduction of tobacco smoking, others characterize them as products that could
undermine efforts to denormalize tobacco use (see online: https://apps.who.int/gb/fctc/PDF/cop6/FCTC_COP6_10-en.pdf). I addition, according to WHO (World Health Organization), most ENDS products have not been tested by independent scientists but the limited testing has
revealed wide variations in the nature of the toxicity of contents and emissions.

Through a language Python program, the authors have analyzed posts excluding private messages or non-public information and inquiring other information such as the constituency building and policy substitution’s targets.

Tobacco companies seek out young people who have significant numbers of followers online and pay them to post photos featuring tobacco brands, or influencers are instructed to include specific hashtags promoting cigarettes on social media posts. Their main targets are American youth.

Just recently, the FTC’s (Federal Trade Commission) Bureau of Competition challenged Facebook’s actions to entrench and maintain its monopoly alleging that the company is illegally denying consumers the benefits of competition (see online: https://www.ftc.gov/news-events/press-releases/2020/12/ftc-sues-facebook-illegal-monopolization).

Inquired posts by authors are a total of 177. See lines 115-143.

A double check is also applied by authors to examine whether publicly available Facebook profile metadata from users who posted messages to ascertain if any had employment or affiliations with the tobacco/ENDS industry.

At line 159-184 authors show results, indicating four main types of political interference: (i) information, (ii) policy substitution, (iii) constituency building, and (iv) legal.

Table 2. at line 185 shows the major thematic areas of political interference in Facebook posts.

Other descriptive statistics about each political influence theme on Facebook pages are shown at line 186-196.

The authors’ conclusions at lines 332-348 are towards a need for future digital surveillance research that characterizes messaging strategies between tobacco/ENDS trade associations and lobbying organizations in the digital sphere.

CHANGE REQUEST

  • At lines 171-72 I would number the detected four main types of political interference in Facebook posts: (i) information, (ii) policy substitution, (iii) constituency building, and (iv) legal,
  • Which are the observed questionable representations of ENDS-related studies and research mentioned in the abstract at lines 26-27? Please clear this point,
  • What do the four main types of political interference: (i) information, (ii) policy substitution, (iii) constituency building, and (iv) legal, mean?

For a people not knowing the California legislation on Tobacco issues, what would be interesting to learn about? Please clear this point.

Kind Regards,

References:

McCausland, K., Maycock, B., Leaver, T., & Jancey, J. (2019). The Messages Presented in Electronic Cigarette-Related Social Media Promotions and Discussion: Scoping Review. Journal of medical Internet research, 21(2), e11953. https://doi.org/10.2196/11953

National Center for Chronic Disease Prevention and Health Promotion (US) Office on Smoking and Health. Preventing Tobacco Use Among Youth and Young Adults: A Report of the Surgeon General. Atlanta (GA): Centers for Disease Control and Prevention (US); 2012. 5, The Tobacco Industry’s Influences on the Use of Tobacco Among Youth. Available from: https://www.ncbi.nlm.nih.gov/books/NBK99238/

Ulucanlar S, Fooks GJ, Gilmore AB (2016) The Policy Dystopia Model: An Interpretive Analysis of Tobacco Industry Political Activity. PLoS Med 13(9): e1002125. https://doi.org/10.1371/journal.pmed.1002125

Author Response

Reviewer Comments and Authors' Point-by-Point Response:

Overall Suggestions:

The purpose of this article is to observe interactions involving the California ENDS (Electronic Nicotine Delivery Systems) industry - including manufacturers, trade associations, retailers, and vaping product users. To this purpose the authors have inquired Facebook pages are online by consumers associations in California. See lines 86-112.

ENDS/ENNDS (Electronic Non-Nicotine Delivery Systems) and cigarettes are substitutes, with higher cigarette prices being associated with increased ENDS/ENNDS sales. Therefore, differential tax policies based on product type could lead to substitution between different types of ENDS/ENNDS and between ENDS/ENNDS and cigarettes (see online: https://escholarship.org/content/qt2f65f2j5/qt2f65f2j5.pdf)..

ENDS are the subject of a public health dispute among bona fide tobacco-control advocates that has become more divisive as their use has increased. Whereas some experts welcome ENDS as a pathway to the reduction of tobacco smoking, others characterize them as products that could undermine efforts to denormalize tobacco use (see online: https://apps.who.int/gb/fctc/PDF/cop6/FCTC_COP6_10-en.pdf). I addition, according to WHO (World Health Organization), most ENDS products have not been tested by independent scientists but the limited testing has revealed wide variations in the nature of the toxicity of contents and emissions.

Through a language Python program, the authors have analyzed posts excluding private messages or non-public information and inquiring other information such as the constituency building and policy substitution’s targets.

Tobacco companies seek out young people who have significant numbers of followers online and pay them to post photos featuring tobacco brands, or influencers are instructed to include specific hashtags promoting cigarettes on social media posts. Their main targets are American youth.

Just recently, the FTC’s (Federal Trade Commission) Bureau of Competition challenged Facebook’s actions to entrench and maintain its monopoly alleging that the company is illegally denying consumers the benefits of competition (see online: https://www.ftc.gov/news-events/press-releases/2020/12/ftc-sues-facebook-illegal-monopolization).

Inquired posts by authors are a total of 177. See lines 115-143.

A double check is also applied by authors to examine whether publicly available Facebook profile metadata from users who posted messages to ascertain if any had employment or affiliations with the tobacco/ENDS industry.

At line 159-184 authors show results, indicating four main types of political interference: (i) information, (ii) policy substitution, (iii) constituency building, and (iv) legal.

Table 2. at line 185 shows the major thematic areas of political interference in Facebook posts.

Other descriptive statistics about each political influence theme on Facebook pages are shown at line 186-196.

The authors’ conclusions at lines 332-348 are towards a need for future digital surveillance research that characterizes messaging strategies between tobacco/ENDS trade associations and lobbying organizations in the digital sphere.

AUTHOR RESPONSE:  We thank the reviewer for their careful review of our manuscript, important additional background information, and other helpful suggestions. In response, we have added additional language and also added certain references in the revised “Introduction” section incorporating reviewer comments. Please see our revised language below:

 “The growing popularity of electronic nicotine delivery systems (ENDS) has resulted in the expansion of the tobacco industry and new concerns about aggressive marketing, underage and young adult uptake, and the growing presence of industry interference (i.e., various tactics used to defeat, weaken or delay public health measures meant to reduce tobacco use) in state and Federal tobacco control policies [1–9]. The emergence of ENDS, including different types of e-cigarettes and other emerging and alternative products (e.g., Heated Tobacco Products (HTPs)) along with their differential pricing, represent new market entries with their own unique health risks, different channels of product marketing and sales, and the presence of new and old tobacco industry actors who promote their use and advocate for their protection from public policymaking[10-13] However, there have been signs of a slowdown in ENDS sales, emphasizing the importance of continuing to strengthen state and Federal tobacco control efforts that have focused on instituting new tobacco/ENDS product taxes, raising the minimum age of sale, and sales bans on certain products (e.g. flavored ENDS products), all aimed at reducing uptake and appeal [8,14].”  [p.1, ¶Introduction 1]

“Facebook, as one of the world’s most popular social media platforms, now serves more than 2.7 billion monthly active users globally, and has been used for political mobilization [16–19]. As a global social networking site, Facebook allows individual users and organizations to create profiles and group pages, share information, send event invitations, communicate through open and direct messages, and has a number of embedded web and mobile-based applications. The combination of these functions makes Facebook a powerful platform for gathering communities, market products using influencers to promote tobacco brands and lifestyles (though Facebook now restricts sponsored ads and influencers for tobacco products), and also represents an important tool for the tobacco/ENDS industry to activate and mobilize opposition to tobacco control laws [20–23].” [p.2, ¶Introduction 3]

Change Request:

  1. At lines 171-172 I would number the detected four main types of political interference in Facebook posts: (i) information, (ii) policy substitution, (iii) constituency building, and (iv) legal.

AUTHOR RESPONSE:  We thank the reviewer for this important suggestion, in response the revised manuscript now has the four main detected types of political interference numbered as suggested.  Please see our revised language below:

“A total of 522 (74.7%) posts were categorized as a form of political interference based on Savell et al. political interference categories, with 339 posts (64.9% of all identified posts) from CCASAA and 183 posts (35.1%) from NC-SFATA. Based on our qualitative analysis, four main types of political interference were detected: (i) information, (ii) policy substitution, (iii) constituency building, and (iv) legal. Among these, there were four main inductively derived categories of political interference: (1) providing updates on e-cigarette-related policy at the federal, state, and local levels (example: “Heads up, Belmont, CA! Flavor ban being considered September 25th!”; meeting the criteria for “constituent building”); (2) sharing opinions about e-cigarette-related policies (example: “Mike Males, senior researcher for the Center on Juvenile and Criminal Justice, drops a fact bomb on the effectiveness of the Tobacco 21 law in California”; meeting the criteria for “policy substitution”); (3) posts related to scientific information related to vaping (example: “There's NO science supporting OUTDOOR bans & misleads public about vaping risks!”; meeting the criteria for “information”); and (4) calls to action on tobacco/e-cigarette policies (example: “OAKLAND, CA: (CALL TO ACTION!) Stop a TOTAL FLAVOR BAN!”]; meeting criteria for “information”). Additional example posts for each of these political interference categories are included in Table 2.”  [p.4, ¶Results1]

  1. Which are the observed questionable representations of ENDS-related studies and research mentioned in the abstract at lines 26-27? Please clear this point.

AUTHOR RESPONSE:  We thank the reviewer for this comment and agree that the observed questionable representations of ENDS- related studies and research is an important result that should be clarified.  In response, we have added addition explanation and relevant citations in the revised “Discussion” section detailing these results. See additional language below:

“More specifically, operators of these industry-driven Facebook community groups and public pages used several digital strategies in an attempt to influence public perception and coordinate efforts of their online constituents. These activities included acting as an online information source about tobacco control legislative developments (including for CA local ordinances, state legislation/referendum, and Federal legislation), inserting discourse and opinion against tobacco control policy with information about policy, and misrepresenting scientific information about the potential harms of ENDS and vaping.  For example, posts observed included those arguing against a study that indicates that chemicals found in e-cigarettes disrupts the gut barrier and triggers inflammation in the body leading to potential harm [42]; and they also questioned the veracity of the methodology of studies that detected potentially harmful chemicals in e-cigarettes[43] . Further, these community groups make specific calls to action to mobilize against tobacco control policies (e.g., inviting online users to political mobilization events, asking them to take direct action against tobacco control policies, etc.).  Many of these strategies overlapped in the text of a single post, and again, originated from users with clear industry affiliations.”  [p.14, ¶1] 

  1. What do the four main types of political interference: (i) information, (ii) policy substitution, (iii) constituency building, and (iv) legal, mean?

AUTHOR RESPONSE:  We thank the reviewer for this important comment and request for addition descriptive language for the four main types of political interference. In response, we added additional language to explain the definition and description of the four main types of political interference: (i) information, (ii) policy substitution, (iii) constituency building, and (iv) legal in our “Results-Content analysis” sectionPlease see our revised language below

3.1. Content analysis

A total of 522 (74.7%) posts were categorized as a form of political interference based on Savell et al. political interference categories, with 339 posts (64.9% of all identified posts) from CCASAA and 183 posts (35.1%) from NC-SFATA. Based on our qualitative analysis, four main types of political interference were detected: (i) information - providing or spreading false or misleading information by understating the health benefits of a proposed policy and overstating its social and economic consequences, (ii) policy substitution- providing policy subsidies, where alternative policies are developed as a substitute for proposed policies, (iii) constituency building- gaining the support of other sectors, organizations, or individuals in order to give the impression of a larger support base for the industry position, and (iv) legal- using or threatening legal action against proposed policies [40]. Among these, there were four main inductively derived categories of political interference: (1) providing updates on e-cigarette-related policy at the federal, state, and local levels (example: “Heads up, Belmont, CA! Flavor ban being considered September 25th!”; meeting the criteria for “constituent building”); (2) sharing opinions about e-cigarette-related policies (example: “Mike Males, senior researcher for the Center on Juvenile and Criminal Justice, drops a fact bomb on the effectiveness of the Tobacco 21 law in California”; meeting the criteria for “policy substitution”); (3) posts related to scientific information related to vaping (example: “There's NO science supporting OUTDOOR bans & misleads public about vaping risks!”; meeting the criteria for “information”); and (4) calls to action on tobacco/e-cigarette policies (example: “OAKLAND, CA: (CALL TO ACTION!) Stop a TOTAL FLAVOR BAN!”]; meeting criteria for “information”). Additional example posts for each of these political interference categories are included in Table 2.”  [p.5, ¶Results 3.1-1]

  1. For a people not knowing the California legislation on Tobacco issues, what would be interesting to learn about? Please clear this point.

AUTHOR RESPONSE:  We thank the reviewer for this helpful suggestion to provide the general readership more context about the uniqueness of CA legislation on tobacco and what would be further interesting to learn about.  In response, we have added additional language to the revised “Discussion” as suggested.  Please see our revised language below:

“In response to this strategic use of Facebook by the ENDS industry, public health stakeholders should expand their own efforts to mitigate and counter pro-vaping narratives, particularly if they misrepresent or include misinformation about tobacco control legislation [44–46]. Importantly, in this study we did not observe any pro-public health or fact-checking of these claims by other Facebook users interacting with these pages or their posts.  Hence, California public health and a tobacco control advocates should seek to directly engage with these sources of information and their audiences on social media platforms such as Facebook, with the aim or better understanding the rationale of pro-ENDS constituents, anticipating arguments against policy implementation that may lack scientific merit, and mobilizing their own public health stakeholders to counter events driven by industry mobilization online. These efforts are particularly important given the uniqueness of the California tobacco control regulatory landscape, with the state leading the way nationally on many progressive policies such as indoor and outdoor smoking bans, efforts to remove illegal and counterfeit vaping products from stores, and a state-wide ban on flavored tobacco and ENDS products that will be decided in a ballot referendum in 2022.”  [p.14, ¶1]

  1. Recommended references:

McCausland, K., Maycock, B., Leaver, T., & Jancey, J. (2019). The Messages Presented in Electronic Cigarette-Related Social Media Promotions and Discussion: Scoping Review. Journal of medical Internet research21(2), e11953. https://doi.org/10.2196/11953

National Center for Chronic Disease Prevention and Health Promotion (US) Office on Smoking and Health. Preventing Tobacco Use Among Youth and Young Adults: A Report of the Surgeon General. Atlanta (GA): Centers for Disease Control and Prevention (US); 2012. 5, The Tobacco Industry’s Influences on the Use of Tobacco Among Youth. Available from: https://www.ncbi.nlm.nih.gov/books/NBK99238/

Ulucanlar S, Fooks GJ, Gilmore AB (2016) The Policy Dystopia Model: An Interpretive Analysis of Tobacco Industry Political Activity. PLoS Med 13(9): e1002125. https://doi.org/10.1371/journal.pmed.1002125

AUTHOR RESPONSE:  We thank reviewer for these important references and have incorporated them into our revised “Introduction” section. 

We thank you again for the opportunity to submit this revised manuscript and look forward to your comments.

Authors